# Association between Statin Use and Diabetes Risk in Patients with Transient Ischemic Attack

**DOI:** 10.3390/ijerph192113770

**Published:** 2022-10-23

**Authors:** Fu-Jun Chen, Ming-Chien Yin, Pei-Yun Chen, Min-Hua Lin, Yi-Hao Peng, Wen-Chao Ho, Pau-Chung Chen, Chung Y. Hsu

**Affiliations:** 1Department of Public Health, China Medical University, No. 91, Hsueh-Shih Road, Taichung 404327, Taiwan; 2Centers for Disease Control Ministry of Health and Welfare, Taichung 40855, Taiwan; 3Division of Respiratory Therapy, China Medical University Hospital, Taichung 404327, Taiwan; 4Department of Nursing, Jen-Teh Junior College of Medicine, Nursing and Management, Miaoli 35664, Taiwan; 5Department of Dietetics, Yunlin Christian Hospital, Yunlin 64866, Taiwan; 6Department of Nutrition, China Medical University, Taichung 404327, Taiwan; 7Department of Respiratory Therapy, Asia University Hospital, Asia University, Taichung 41354, Taiwan; 8Department of Nursing, Asia University, Taichung 41354, Taiwan; 9Department of Public Health, National Taiwan University, Taipei 10617, Taiwan; 10National Institute of Environmental Health Sciences, Miaoli 35053, Taiwan; 11Graduate Institute of Biomedical Sciences, China Medical University, Taichung 404327, Taiwan

**Keywords:** cohort study, diabetes mellitus, transient ischemic attack, statins

## Abstract

Statin therapy can effectively reduce recurrent transient ischemic attack (TIA) risk. However, studies have reported that statin use is associated with incidence of diabetes mellitus (DM). Whether statin therapy remains associated with higher DM risk in patients with TIA remains unknown. This study investigated whether statin treatment influences incident DM risk in patients with TIA. We conducted a retrospective cohort study using the Longitudinal Health Insurance Database 2000. Participants who were newly diagnosed with TIA (ICD-9-CM code 435) from 1 January 1997 to 31 December 2011 were recruited. The Kaplan–Meier method and Cox proportional risk model of time-dependent covariance were used. We enrolled 8342 patients with newly diagnosed TIA from 1 January 1997 to 31 December 2011. Of these, 1255 patients were classified as statin users and 7087 as nonusers. During the 14-year follow-up, the incidence of newly diagnosed DM was 0.545-fold lower in the statins group compared with nonusers (95% confidence interval [CI] = 0.457–0.650). According to cumulative defined daily doses (cDDDs), the adjusted hazard ratios for DM were 0.689, 0.594, and 0.463 when patients were treated with statins at cDDDs = 28–89, 90–180, and >180, respectively. In patients with TIA, statin use is associated with a lower incident DM risk compared with the nonuse of statins.

## 1. Introduction

Transient ischemic attack (TIA) is a common neurological disease. For patients exhibiting an acute loss of focal cerebral or ocular function attributable to vascular disease lasting under 24 h, TIA should be considered as the diagnosis [1]. TIA is a clinically important predictor of ischemic stroke. Approximately 13% of patients experience TIA prior to stroke [2]. TIA increases the risk of stroke to 5% within 48 h and 10% within 3 months. Therefore, adequate treatment of TIA is necessary to reduce the risk of stroke [2,3]. 

Atherosclerosis is a common risk factor for TIA. Patients with TIA caused by atherosclerosis should receive statin therapy [4]. Statins are a mainstay treatment for hyperlipidemia and atherosclerosis cardiovascular disease [5] and are recommended by the 2013 ACC/AHA guidelines for patients with ischemic stroke or TIA as a secondary prevention strategy [6]. Good adherence to statin therapy is associated with superior clinical outcomes for recurrent ischemic stroke, hemorrhagic stroke, and acute coronary events [7]. However, some studies report that the effect of statin therapy is different for older and middle-aged adults. Although statins are effective for primary prevention of atherosclerotic cardiovascular disease in patients aged 40 to 75 years, statins are potentially harmful to patients older than 75 years [8]. 

Statin use has been associated with increased insulin resistance and has been reported to elevate the risk of incident diabetes mellitus (DM) [9,10]. A retrospective matched-cohort study that investigated the relationship between statin initiation and diabetes progression indicated that DM occurred in 55.9% of statin users and 48.0% of comparators (odds ratio, 1.37; 95% confidence interval [CI], 1.35–1.40; *p* < 0.001) [10]. The relationship between statin use and DM may be attributable to reductions in insulin sensitivity and insulin secretion [11]. One meta-analysis evaluated the relationship between statin use and new-onset DM. Cohort and case–control studies were selected. The results demonstrated that new-onset DM risk was higher in statin users than in nonusers (relative risk 1.44; 95% CI 1.31–1.58) [12].

Research indicates that statin use contributes to the development of DM; however, the results of such studies vary widely. The risk of new-onset DM with statin use is between 0.1% [13] and 46% [11]. Some studies have reported conflicting results and propose that statin use reduces the risk of DM. One study recruited 5974 participants aged 45–64 years to investigate the association between statin use and DM using a randomized placebo-controlled trial of 40 mg pravastatin. The results indicated a 30% reduction in the incidence of new-onset DM [14]. 

Whether statin treatment remains associated with increased DM risk in patients with specific health conditions, such as TIA, is unknown. The determination of the effect of statins on incident DM in patients with TIA will allow clinicians to optimize treatment plans.

This study investigated whether statin treatment influences incident DM risk in patients with TIA using data from the Longitudinal Health Insurance Database 2000 (LHID 2000).

## 2. Materials and Methods

### 2.1. Study Population 

In Taiwan, the National Health Insurance (NHI) program has been a compulsory, single-payer program since 1995, and more than 99% of the population is insured. The National Health Insurance Research Database (NHIRD) contains patient registry and claims data, including demographic characteristics, details of outpatient visits, hospitalizations, prescriptions, and diagnoses. The LHID 2000 is a subset of the NHIRD that contains reimbursement claims data of one million individuals randomly selected from the NHIRD. The NHI reported no difference in the sex and age distributions of participants in the LHID 2000 and NHIRD.

### 2.2. Study Participants and Exposure Ascertainment 

This study used a retrospective cohort design. The sample was enrolled from the LHID 2000, from between 1 January 1997 and 31 December 2011. The definition of diseases was based on the International Classification of Disease, Ninth Revision, Clinical Modification (ICD-9-CM). To further validate the diagnoses of the study, only patients with at least three outpatient visits or who had ever been hospitalized because of their diagnosis were selected.

Patients who were newly diagnosed with TIA (ICD-9-CM code 435) from 1 January 1997 to 31 December 2011 were included. The exclusion criteria were (1) a TIA or DM (ICD-9-CM code 250) diagnosis before 1 January 1997, (2) diabetic medical records before TIA diagnosis, (3) DM or stroke within 1 year following TIA diagnosis, and (4) statin medication records before the diagnosis of DM. 

The comorbidities of participants included hypertension (ICD-9-CM codes 401–405), dyslipidemia (ICD-9-CM code 272), atrial fibrillation (ICD-9-CM code 427.31), heart failure (ICD-9-CM code 428), coronary artery disease (ICD-9-CM codes 410–414), peripheral artery disease (ICD-9-CM code 443.9), chronic kidney disease (ICD-9-CM code 585.9), chronic obstructive pulmonary disease (ICD-9-CM codes 491 and 492), and alcohol-related diseases (291, 303.0, 303.9, 305.0, and 571.0–571.3). A flow chart of participant enrollment is presented in Figure 1.

Data relating to patient statin prescriptions were obtained from the medication records in the LHID 2000. Both hydrophilic and lipophilic statins were selected in the study, including pravastatin, fluvastatin, rosuvastatin, simvastatin, lovastatin, and atorvastatin. Exposure to statins was measured in terms of defined daily dose (DDD) as the unit of calculation, and the daily dose of each prescribed medicine was calculated using the following formula:$$\left(\mathrm{Number}\;\mathrm{of}\;\mathrm{DDDs}\right)\;=\;\frac{\left(\mathrm{Total}\;\mathrm{amount}\;\mathrm{of}\;\mathrm{drug}\right)}{\left(\mathrm{Amount}\;\mathrm{of}\;\mathrm{drug}\;\mathrm{in}\;a\;\mathrm{DDD}\right)}$$

To consider long-term exposure to changes in the exposure dose of each drug, we calculated the cumulative DDDs (cDDDs) for each participant under the tracking time of the study as the drug exposure dose of the individual. Statin exposure was grouped according to the cumulative definition of statins used as the previous research [15]. The nonexposed group was defined as having cDDD < 28. The exposed group was categorized into low-, moderate-, and high-use groups (cDDD = 28–89, 90–180, and >180, respectively).

For descriptive statistics, Student’s t-test and chi-square tests were conducted to identify differences in demographic characteristics between the statin use and statin nonuse groups. The Kaplan–Meier (KM) method and Cox proportional risk model of time-dependent covariance were used to evaluate the association between statin use and incident DM in patients with TIA. Hazard ratios (HRs) and 95% confidence intervals were calculated to quantify DM risk. Statistical analyses were performed using SAS (version 9.4; SAS Institute, Cary, NC, USA), and a two-tailed *p* < 0.05 was considered statistically significant.

## 3. Results

Patients (N = 8342) with newly diagnosed TIA, between 1997 and 2011, were selected for the study. Of these, 1255 were statin users and 7087 were statin nonusers. The mean age at TIA diagnosis was significantly lower for statin users than for statin nonusers (61.90 ± 11.70 vs. 63.20 ± 15.20, *p* < 0.0001). No differences in TIA diagnosis between male and female patients were observed (*p* = 0.3615). Approximately half of the participants were male (49.45%). The mean years of follow-up were 7.5 versus 6.1 for statin users and statin nonusers, respectively (see Table 1). 

The incidence rate of DM during follow-up between 1997 and 2011 was 14.98% in statin users and 14.07% in nonusers. Rates of hypertension, dyslipidemia, coronary artery disease, peripheral artery disease, and chronic kidney disease were significantly higher in statin users than in statin nonusers (Table 1). Among statin users, 48.53%, 20.96%, and 30.51% were in the high-use (cDDD > 180), moderate-use (cDDD = 90–180), and low-use (cDDD < 28) groups, respectively. The use of prescriptions of aspirin, nonsteroidal anti-inflammatory drugs, angiotensin-converting enzyme inhibitors, angiotensin II receptor blockers, beta blockers, calcium channel blockers, diuretics, nonstatin lipid-lowering drugs, and fibrates was significantly higher in statin users compared with nonusers.

Table 2 exhibits the HRs of incident DM in statin users and nonusers with TIA. Users were at a significantly lower DM risk than nonusers (adjusted HR = 0.545, 95% CI = 0.457–0.650) after adjustments for age, sex, income, degree of urbanization, comorbidities, and medication use. Regardless of the cDDDs of the statins, incident DM risk in patients with TIA was significantly lower in all three groups of statin users compared with nonusers. Adjusted HR for DM ranged from 0.463 to 0.689 was the highest for the low-use group (28–89 cDDDs) and the lowest for the high-use group (>180 cDDDs). 

Figure 2 presents estimates cumulative incidence of new-onset DM during follow-up using the KM method. Regardless of statin-user or statin-nonuser group, the incidence of type 2 diabetes increased by age [16]. The results indicate that the cumulative incidence of DM in the use of the statins group was lower than that in the statin-nonuse group in the early stages of the follow-up. At approximately 12 years of follow-up, the incidence of DM in the statins group was higher than that of the statin-nonuse group. A significant difference in the log-rank test (*p* = 0.032) was observed. The results indicated that the incidence of DM in TIA patients was lower in the statin-use group than in the non-statin-use group during the first few years of statin use. However, the slope of KM curve was steeper in the statin-use group than in the non-statin group, which means the incidence of DM in the statin-use group increased faster than that of the non-statin group in the TIA patient. The incidence of DM in patients using statin in the TIA patient was lower than that of non-statin patients during the initial tracking time. However, if the tracking time was longer, the results were the opposite; in particular, when the tracking time was longer than 12 years, the incidence of DM was higher in statin nonusers than in statin users.

The time-dependent covariate Cox model results are displayed in Table 3. Female sex, age, menopause, hypertension, and lack of hyperlipidemia did not affect the results of the model. The effects of statins on first-time DM among patients with TIA is significantly associated with several risk factors, including drug use, male sex, hyperlipidemia, and follow-up time (<9.2 years). Significant effects of statin on incident DM among patients with TIA were observed only during follow-up periods under 9.2 years. Statin use was associated with a reduced risk of DM when the tracking time was less than 9.2 years. The lowest HRs for suffering DM was the 90–180 cDDDs group. Other risk factors, including ARB, aspirin, NSAIDs, gender, hypertension, and hyperlipidemia, significant influenced the HR of developing DM in TIA patients. 

Compared with Figure 2, when the follow-up period is less than 12 years, the incidence of DM in non-statin users is higher than that in statin users. Table 3 showed the results of the time-dependent covariate Cox model analysis that the HR of DM in the TIA patient using statin drugs was only significant for those tracking times was less than 9.2 years. The incidence of DM was higher in non-statin users than statin users during the tracking time less than 9.2 years. 

## 4. Discussion

We observed that when the statin-use duration was less than 9.2 years, in patients with TIA, statin use was associated with a lower incident DM risk (adjusted HR = 0.545), and DM risk substantially decreased with an increase in the cDDDs of statins. Previous studies indicated that statin users exhibited a higher DM risk than statin nonusers [10,12]. This is inconsistent with our findings. The participants recruited for our research were all patients who were newly diagnosed with TIA. The participants’ characteristics differed from those of participants of previous studies. 

We analyzed cohort data using a Cox proportional risk model of time-dependent covariance that is suitable for nonconstant exposure, such as nonconstant drug use, air pollution exposure, and temperature exposure. The drug dose that patients receive is adjusted according to their clinical situation, especially in Taiwan, where the health insurance system is convenient for the population. Therefore, the Cox proportional risk model of time-dependent covariance is suitable for situations where exposure is not constant. Some studies have reported decreasing trends in the incidence of DM for statin users [14], but other studies have reported the opposite results [10,12]. Differences in the results of our study with those of other studies may be because of differences in the statistical methods used. The majority of our sample was aged over 65 years (49.31%). A previous study indicated that the effect of statin therapy is different for older and middle-aged adults [8], and therefore, differences in age distribution might contribute to the divergent results. 

The previous literature indicated that statin use has been associated with increased insulin resistance and has been reported to elevate the risk of incident DM [9,10]. However, there were also a lot of risk factors associated with DM, such as age [16], body weight, and serum uric acid [17]. The interference related to DM demonstrated that the results about incident DM with TIA were diverse. This might contribute to the fact that our results were inconsistent with some literature. 

The previous literature indicated that the incidence rate of the onset of type 2 diabetes increased by age [16]. The KM method indicated that the incidence of DM increased with the tracking time. These results were consistent with the previous literatures. The cumulative incidence of DM in the statin use group was lower than that in the statin nonuse group in early stages of the follow-up period. However, when the cohort was followed for longer than 12 years, the converse result was observed. These results may be influenced by the length of the follow-up period. This is potentially why our results are inconsistent with some of those reported in previous studies. It is shown that the risk of developing DM after using statin drugs in TIA patients is diverse, and that it is dependent on the length of tracking time. 

One study reported that statin use is beneficial for cardiovascular disease prevention [18]. This is inconsistent with our findings. Our study indicated that hypertension, dyslipidemia, and coronary artery disease prevalence was significantly higher in statin users than in statin nonusers. Inconsistency in the prevalence of cardiovascular disease between our results and those of prior studies may be because of the age of participants. The majority of the participants (49.31%) in our study were older than 65 years. The literature indicates that uncertain harms should be considered when patients are older than 75 years [8]. 

The exposure to statins, in terms of dose, does not remain constant clinically. The metabolism and absorption of each drug vary in different situations. Because drug dose is not constant and drug absorption is different for different people, research that combines big data and laboratory data should be undertaken in the future.

This study has the following limitations. First, by using the LHID 2000, we defined the situations of patients only on the basis of ICD-9-CM code entered upon treatment in a hospital or clinic. Some information was lacking in the health insurance database, such as health examination data (such as serum and biochemical data), daily life habits (such as smoking and drinking), family history (such as hereditary disease), and life environment (such as exposure to pollution). Because of the limited information available, potential confounders that might have influenced the relationship between statin use and DM could not be included in analysis. Second, the accuracy of the statin dosages might have been somewhat less than optimal, although we calculated the cumulative dose using a credible formula; nevertheless, compliance with the doctor’s orders for each patient could not be confirmed. Finally, we only recruited patients living in Taiwan. Some discrepancies in the incidence of diseases between ethnicities have been reported. Research using a multiethnic sample should be considered in the future.

## 5. Conclusions

In patients with TIA, when the statin-use duration is less than 12 years, statin use is associated with a lower DM risk compared with the nonuse of statins. However, when the cohort was followed for longer than 12 years, the converse result was observed. It is important that the follow-up period in statin use contributes to determining whether there is a risk of development of DM in patients with TIA. 

## 6. Limitation

In Taiwan, if cardiovascular disease patients have TC ≥ 160 or LDL-C ≤ 100, statin use is eligible through Taiwan’s health insurance. Those patients might not be diagnosed as dyslipidemia. Therefore, there will be a gap between the data of dyslipidemia-diagnosed and statin-using patients. 

The purpose of our research was to investigate whether statin treatment influences incident DM risk in patients with TIA. Therefore, we defined “statin treatment” via “the cases who actually use statin-drugs”.

## Figures and Tables

**Figure 1 ijerph-19-13770-f001:**
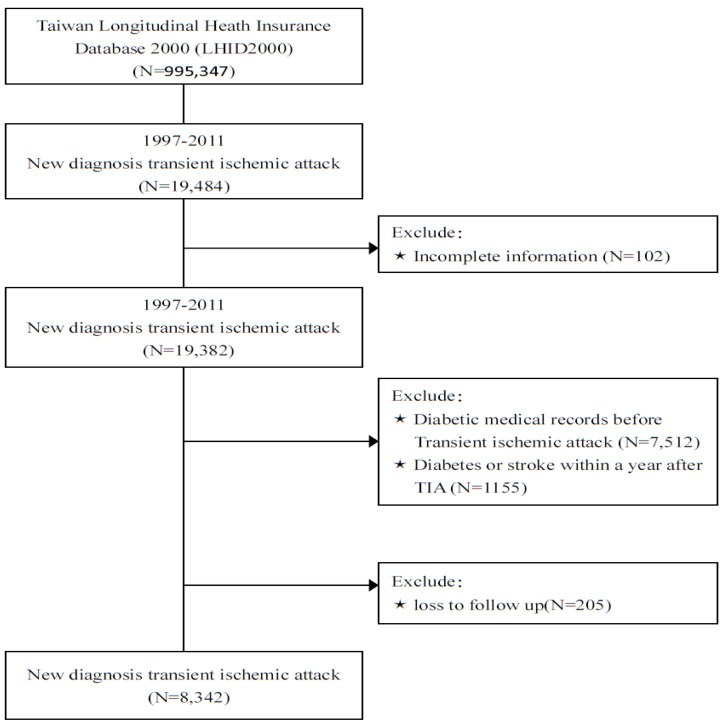
Flow chart for the enrollment of participants in the study.

**Figure 2 ijerph-19-13770-f002:**
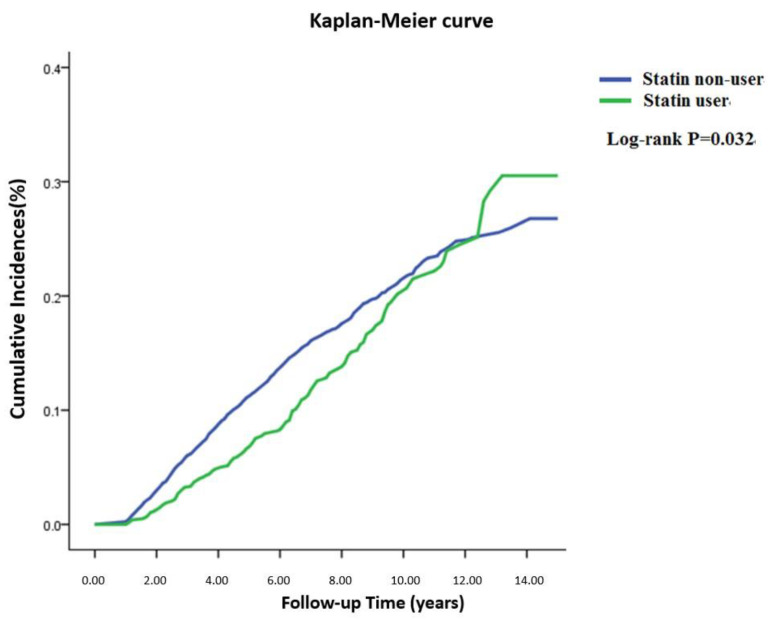
Cumulative incidence of diabetes mellitus by statin use among TIA patients in Taiwan during follow-up time.

**Table 1 ijerph-19-13770-t001:** Baseline demographics of patients with transient ischemic attack.

	All TIA		Statin User		Non-Statin User		
	(N = 8342)		(N = 1255; 15.1%)		(N= 7087; 84.9%)		
Variable	No.	%	No.	%	No.	%	*p* Value
Age of TIA onset †							<0.0001
<45	1029	12.34	100	7.97	929	13.11	
45–65	3199	38.35	615	49.00	2584	36.46	
>65	4114	49.31	540	43.03	3574	50.43	
Med (IQR)	14.6 (52.70–74.40)		62.7 (53.20–70.90)		65.2 (52.60–75.10)		
Mean age ± SD	63.0 ± 14.80		61.90 ± 11.70		63.20 ± 15.20		
Male sex	4167	49.45	612	48.76	3555	50.16	0.3615
Insurance premium †, NT$							<0.0001
0	2547	30.53	448	35.70	2099	29.62	
1–15,840	1770	21.22	231	18.41	1539	21.72	
15,841–25,000	2817	33.77	379	30.20	2438	34.40	
≥25,000	1208	14.48	197	15.70	1011	14.27	
Incident DM Comorbidity	1185	14.21	188	14.98	997	14.07	
Hypertension	6596	70.62	1075	85.66	4855	68.51	<0.0001
Dyslipidemia	3402	36.42	1097	87.41	2034	28.70	<0.0001
Atrial fibrillation	1215	13.01	148	11.79	950	13.40	0.1195
Heart failure	1746	18.69	249	19.84	1334	18.82	0.3969
Coronary artery disease	4527	48.47	758	60.40	3354	47.33	<0.0001
Peripheral arterial disease	750	8.03	140	11.16	557	7.86	0.0001
Chronic kidney disease	869	9.30	143	11.39	622	8.78	0.0031
COPD	2653	28.40	363	28.92	2027	28.60	0.8158
ARD	224	2.40	19	1.51	167	2.36	0.0624
Statin dose †							<0.0001
<28 cDDDs			0	0.00	7087	100.00	
28–90 cDDDs			383	30.51	0	0.00	
90–180 cDDDs			263	20.96	0	0.00	
>180 cDDDs			609	48.53	0	0.00	
Aspirin	4180	50.11	900	71.71	3280	46.28	<0.0001
NSAIDs	6146	73.68	1049	83.59	5097	71.92	<0.0001
Antihypertensive agent							
ACEi	2304	17.62	519	41.35	1785	25,19	<0.0001
ARB	2438	29.23	592	47.17	1846	26.05	<0.0001
Beta blocker	2973	35.64	688	54.82	2285	32.24	<0.0001
CCB	4214	50.52	868	69.16	3346	47.21	<0.0001
Diuretic	2850	34.16	558	44.46	2292	32.34	<0.0001
Antihyperlipidemic agent							
Nonstatin lipid-lowering drug	108	1.29	66	5.26	42	0.59	<0.0001
Fibrate	557	6.68	275	21.91	282	3.98	<0.0001
Follow-up time (years)							
Med (IQR)	5.8 (3.20–9.20)		7.60 (4.70–10.10)		5.50 (2.90–8.90)		
Mean (SD)	6.3 ± 3.60		7.50 ± 3.40		6.1 ± 3.60		

Supplement. Abbreviations: TIA, transient ischemic attack; Med (IQR), median (interquartile range); NT$, New Taiwan Dollar; SD, standard deviation; DM, diabetes mellitus; COPD, chronic obstructive pulmonary disease; ARD, alcohol-related disease; cDDDs, cumulative defined daily doses; NSAID, nonsteroidal anti-inflammatory drug; ACEi, angiotensin-converting enzyme inhibitor; ARB, angiotensin-receptor blocker; CCB, calcium-channel blockers. † Models adjusted for gender, age, income, urbanization, comorbidity, and drug used.

**Table 2 ijerph-19-13770-t002:** Hazard ratios of incident diabetes mellitus between statin users and nonusers with transient ischemic attack.

Variable	Crude Diabetes HR (95% CI)	Adjust Diabetes HR (95% CI)
Statin nonuser (<28 cDDDs)	1.000	1.000
Statin user (≥28 cDDDs) 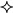	0.843 (0.722–0.986) *	0.545(0.457–0.650) **
Statins dose 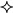		
28–90 cDDDs	0.911 (0.694–1.195)	0.689 (0.520–0.911) **
90–180 cDDDs	0.973 (0.715–1.326)	0.594 (0.431–0.818) **
>180 cDDDs	0.762 (0.615–0.945) *	0.463 (0.367–0.584) **

Supplement. Abbreviations: CI, confidence interval; cDDDs, cumulative defined daily doses; HR, hazard ratio. 
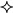
 Statin use and categorization was defined by the cumulative prescription (cDDDs) of statins. Models adjusted for gender, age, income, urbanization, comorbidity, and drug used. * *p* < 0.05, ** *p* < 0.01.

**Table 3 ijerph-19-13770-t003:** Time-dependent covariate Cox model for the effects of statin on first-time diabetes mellitus among transient ischemic attack patients.

	Statin Nonuser	Statin Use Levels *						*p* Trend
		28–89 cDDDs		90–180 cDDDs		>180 cDDDs		
	HR	HR	95% CI	HR	95% CI	HR	95% CI	
ARB	1.0	0.931	0.689–1.257	0.477	0.289–0.786	0.642	0.382–1.078	0.0123
Aspirin	1.0	0.886	0.656–1.197	0.452	0.274–0.745	0.604	0.360–1.013	0.0043
NSAIDs	1.0	0.943	0.699–1.272	0.484	0.294–0.797	0.659	0.393–1.104	0.0159
Subgroup analysis †								
Gender								
Female	1.0	0.805	0.525–1.232	0.658	0.368–1.176	0.731	0.375–1.427	0.3223
Male	1.0	1.146	0.750–1.749	0.271	0.101–0.730	0.572	0.253–1.292	0.0297
Age								
<65	1.0	1.007	0.693–1.463	0.583	0.326–1.043	0.635	0.324–1.243	0.1796
≥65	1.0	0.782	0.471–1.299	0.300	0.111–0.806	0.595	0.264–1.341	0.0503
Menopause								
Yes	1.0	0.878	0.561–1.373	0.653	0.346–1.233	0.777	0.382–1.580	0.5041
No	1.0	0.427	0.103–1.778	0.810	0.194–3.381	0.664	0.090–4.919	0.6711
Follow-up time								
<9.2	1.0	0.593	0.439–0.801	0.552	0.392–0.779	0.480	0.373–0.619	<0.0001
≥9.2	1.0	1.435	0.6520–3.157	1.072	0.440–2.614	0.868	0.471–1.601	0.7175
Hypertension								
Yes	1.0	0.943	0.689–1.292	0.526	0.319–0.868	0.645	0.378–1.102	0.0351
No	1.0	0.718	0.262–1.966	-	-	0.800	0.111–5.758	0.9276
Hyperlipidemia								
Yes	1.0	0.798	0.578–1.101	0.449	0.267–0.753	0.606	0.354–1.038	0.0041
No	1.0	1.418	0.630–3.189	0.360	0.050–2.571	0.472	0.066–3.367	0.5061

Supplement. Abbreviations: cDDDs, cumulative defined daily doses; HR, hazard ratio; 95% CI, 95% confidence interval, angiotensin Ⅱ receptor blocker; NSAIDs, nonsteroidal anti-inflammatory drugs. * Statin use and categorization were defined by cumulative prescription (cDDDs) of statin. † Models adjusted sex, age, income, urbanization, comorbidity, drug use.

## Data Availability

National Health Insurance Research Database, Taiwan. Available online: http://nhird.nhri.org.tw/en/index.htm, accessed on 20 January 2021.

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
