# Peer review of "Association between Statin Use and Diabetes Risk in Patients with Transient Ischemic Attack"

_ijerph, 2022, doi:10.3390/ijerph192113770_

Round 1

Reviewer 1 Report

1.    Since this study aimed to investigate whether statin use will influence diabetes risk in study population, statin use prior to diabetes diagnosis is necessary. The exclusion criteria (4): “statin medication records before the diagnosis of DM” is therefore strange.

2.    There are only 87.41% statin users who are comorbid with dyslipidemia. It means that there are more than ten percent patients using national health insurance paid statins without definite dyslipidemia diagnosis. The accuracy of database is quite suspicious.

3.    The KM method and sensitivity analysis showed converse trend of DM incidence in statin users and non-users but without adequate explanation.

4.    The author concluded that “In patients with TIA, statin use is associated with a lower DM risk compared with the nonuse of statins” according to statistical analysis, but lack of possible mechanism discussion.

Reviewer 2 Report

Thank you for the opportunity to review your manuscript titled “Association Between Statin Use and Diabetes Risk in Patients with Transient Ischemic Attack”. Several problems are required to revise.

1.     Line 48, the word “guideline s” should delete the space.

2.     Line 119-120, “The exposed group was categorized into low-, moderate-, and high-119 use groups (cDDD = 28–89, 90–180, and >180, respectively).” This categorical method is different from the description in Abstract. Which’s method is correct? Additionally, the Line 143-145, the categorical values are different from the previous description. Please re-check and revise it.

3.     Why did the authors use the cDDD = 28–89, 90–180, and >180, to class the groups? Are there any references to support your point?  

4.     Table 1, there were some abbreviations in the Table (e.g., Med; ARD). Please add more explanation for the abbreviation below the Table.

5.     Conclusion: please make the conclusions clearer and more specific to illustrate the results of the study.

6.     The English language should be revised by a native speaker. 

Round 2

Reviewer 1 Report

The author had addressed all comments and suggests accordingly.